# Diffusion-based Data Generation for Out-of-Distribution Object Detection

## Abstract

Generating out-of-distribution (OOD) data is critical for training OOD object detectors, enabling them to identify OOD objects or categories as "unknown". Previous methods may generate imprecise OOD features due to incorrect assumptions on in-distribution (ID) data distribution. In this paper, we propose to discard any distribution assumption, leveraging a diffusion model to faithfully model the ID data distribution, and design a filtering strategy to generate accurate OOD data samples for training an unknown-aware object detector. Unlike previous methods that rely on predefined parametric models for modeling distributions, our diffusion model captures the latent feature distributions of ID data, which allows us to synthesize data samples within a compact feature space. We further design a filtering strategy based on K-Nearest Neighbors (KNN) to select low-density data samples proximate to the ID data as generated OOD samples, which are more challenging and effective for improving the OOD detector. Our method is generic and can be easily integrated with existing baseline methods, demonstrating superior performance on multiple benchmark datasets. The code will be made publicly available.

## 1 Introduction

In real open world scenario, it is essential for an object detector to possess the ability to determine "what it does not know" to ensure reliability and security. However, most existing object detectors (Girshick, 2015; Ren et al., 2015; Wang et al., 2022; He et al., 2017; Liu et al., 2016; Redmon et al., 2016; Lin et al., 2017b;a; Chu et al., 2020; Carion et al., 2020; Wang et al., 2021b; Sun et al., 2021a; Zhu et al., 2020) often fail to handle objects that have not been presented during training and assign them incorrect labels at high confidence (Dhamija et al., 2020). This might lead to serious issues in safety-critical applications. For example, in autonomous driving, a perception system that confuses an unseen, unexpected object as one it has encountered during training could cause a fatal accident. This motivates us to explore out-of-distribution (OOD) object detection, which aims to localize and classify in-distribution (ID) object categories seen during training while simultaneously distinguishing OOD objects that have never been exposed during training from ID objects.

Albeit important, OOD object detection remains a challenging problem due to the inaccessibility of training data for OOD objects. Existing research (Du et al., 2022b) attempts to synthesize OOD data by estimating the ID data distribution in the latent feature space and sampling low-density data as OOD data (Bishop, 1994). While these approaches have shown promising results, they often rely on a pre-defined distribution assumption (Du et al., 2022b;a), such as the Gaussian Mixture Model (GMM). The distribution assumption may not reflect the true data distribution in practice (as in Figure 1, GMM cannot fit the distribution of original data well). As a result, the sampled OOD data may not be representative of the true OOD data and may even conflict with high-density ID data, leading to sub-optimal performance of an OOD object detector.

To address above metioned problem, we introduce OpenDiffusion, which leverages the diffusion model to fit data distributions, generate data samples, and further incorporates a non-parametric filtering scheme to produce meaningful OOD data for optimizing the OOD detector. Firstly, inspired by the success of diffusion models (Ho et al., 2020; Song & Ermon, 2020) on modeling accurate distributions, we train a diffusion model on the instance-level latent features extracted by a pre-trained object detector from ID data. This enables the model to effectively model the ID data distribution in

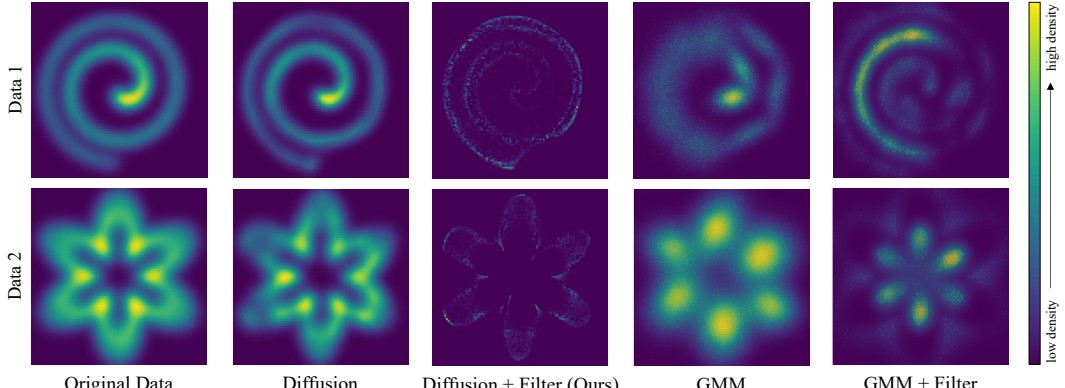

Figure 1: Comparison of the diffusion model and Gaussian Mixture Model (GMM) on fitting two toy distributions. Given the original data, we use the diffusion model and GMM to model the ID data distribution and generate/sample data using the trained model: "Diffusion" and "GMM"; Then, we acquire low-density OOD data using our filtering scheme: "Diffusion + Filter" and "GMM + Filter". It can be observed that the data generated by the diffusion model accurately captures the original ID data distribution. Furthermore, the filtered data effectively represent low-density OOD samples.

the latent feature space and synthesize novel data samples by reversing a diffusion process. Secondly, we design a non-parametric filtering strategy to select synthesized samples that are likely to be OOD samples based on their density under the ID data distribution. To measure the density, we find the K-nearest neighbors (KNN) of each synthesized sample within the ID dataset and calculate the distance to its farthest neighbor. If the distance exceeds a pre-defined threshold, implying low density under the ID data distribution, we regard the sample as an OOD sample. Additionally, the generated samples far away from the ID data are discarded in the latent feature space, as they are either meaningless noise or trivially different from ID data (see Figure 1: dark blue regions away from ID data). Such samples do not benefit the training of the OOD detector and may even hinder the learning process. As illustrated in Figure 1, the diffusion model can faithfully represent the original data distribution, and our sampling strategy obtains meaningful samples of low probability density (OOD) that are close to the ID data manifold (see Figure 1: Diffusion + Filter). Meanwhile, we employ prototype-based representation learning loss to enhance the compactness and separability of the latent feature space for ID data.

Comprehensive experiments conducted on the OOD benchmarks (Everingham et al., 2010; Lin et al., 2014; Yu et al., 2020; Kuznetsova et al., 2020) illustrate the efficacy of our method. Notably, our model outperforms the baseline by more than **7%** on average in terms of FPR95 on multiple datasets and achieves state-of-the-art results. To summarize, our contributions can be listed below:

- To the best of our knowledge, we are the first to employ the diffusion model in generating OOD samples. Without any prior distribution assumption, the diffusion model can estimate the intricate ID data distribution for subsequent OOD data sampling.

- We design a filtering strategy to select OOD samples that have low probability density and are close to the ID data manifold in the latent space. The discriminative decision boundary that discriminates ID and OOD samples can be well learned by fine-tuning the OOD detector on the obtained OOD samples.

- Our approach is capable of generating OOD samples that closely resemble ID data. To the best of our knowledge, this particular aspect has never been considered in prior research. Extensive experiments conducted on diverse datasets(Everingham et al., 2010; Lin et al., 2014; Kuznetsova et al., 2020; Yu et al., 2020) illustrate the efficacy of our approach.

## 2 PRELIMINARY

**Task** We focus on studying OOD object detection to recognize OOD objects as "unknown" while localizing and classifying ID objects. The object detection dataset is denoted as $\mathcal{D}$, comprising the image $\mathbf{x} \in \mathcal{X}$; the ground-truth bounding box coordinates $\mathbf{b} \in \mathcal{B}$ and its semantic label $y \in \mathcal{Y}$. The closed training set consists exclusively of ID categories ($K$ categories) and is denoted as

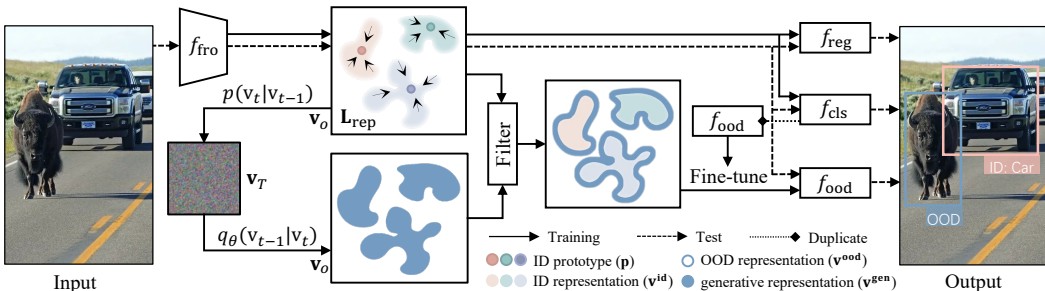

Figure 2: Overview of the proposed method. Only using ID data, we extract proposals and refine the feature space with our representation learning loss ($\mathbf{L}_{\text{rep}}$) during training. We then use a diffusion model to generate new data and a proposed filter to select OOD samples on the low-density region of the ID data manifold. After fine-tuning the OOD detector head ($f_{\text{ood}}$), our network can identify and localize ID/OOD objects from open datasets and classify ID objects (an example image from an open dataset is shown to illustrate the inference).

$\mathcal{D}_{\text{close}} = \left\{ \left( \mathbf{x}^{\text{id}}, \mathbf{b}^{\text{id}}, \mathbf{y}^{\text{id}} \right) \right\}$. During inference, besides identifying ID objects and assigning them with labels from the $K$ categories, the detector should discern whether an object from any dataset $\mathcal{D}_{\text{open}}$ (open set) belongs to an ID category or constitutes an OOD category, which is not part of the $K$ categories introduced during training.

**Baseline**    In this task, we use the well-performing and efficient VOS (Du et al., 2022b) model as our main baseline detector to instantiate our proposed method, and some extended baseline methods and tasks will also be presented. Considering the two-stage object detection framework (Girshick, 2015), for a given input image $\mathbf{x}$, the model generates proposals by the backbone network and RPN and encodes them into a latent feature space. Subsequently, R-CNN classifies and locates these proposals using the classification head and regression head, respectively. For the baseline model that already possesses OOD detection capabilities, each proposal is assigned an OOD score from a classification head to indicate the model's confidence in whether the proposal belongs to the OOD category.

**Diffusion Formulation**    In our approach, we utilize the Denoising Diffusion Probabilistic Model (DDPM) to synthesize Out-of-Distribution (OOD) features, thus we provide a brief overview of its basic concept. Given an unknown target data distribution $p(\mathbf{v}_0)$, DDPMs model the data distribution by learning the inverse of a forward diffusion process, a mechanism that gradually applies noise to data drawn from the target distribution. This forward diffusion process is built with a Gaussian Markov chain and formally represented as $p(\mathbf{v}_t \mid \mathbf{v}_{t-1}) := \mathcal{N}\left(\mathbf{v}_t; \sqrt{1 - \beta_t}\mathbf{v}_{t-1}, \beta_t \mathbf{I}\right)$, where $t$ ranges from 1 to $T$ and $\{\beta_t\}_1^T$ are predetermined hyperparameters. With a large number of timesteps $T$, this diffusion process effectively obfuscates the information in the input samples $\mathbf{v}_0$ such that $p(\mathbf{v}_T) := \mathcal{N}(\mathbf{v}_t; \mathbf{0}, \mathbf{I})$. The objective of training diffusion models is to learn the reverse process, also modeled by a Markov chain: $q_\theta(\mathbf{v}_{t-1} \mid \mathbf{v}_t) = \mathcal{N}(\mathbf{v}_{t-1}; \mu_\theta(\mathbf{v}_t), \Sigma_\theta(\mathbf{v}_t))$, where the statistics are optimized by maximizing the variational lower bounds on the log-likelihood over the training data. More specifically, by reparameterizing with a noise prediction neural network $\epsilon_\theta$, the model can be trained using a simple mean-squared loss (Ho et al., 2020) between the predicted noise $\epsilon_\theta(\mathbf{v}_t, t)$ and the actual sampled Gaussian noise $\epsilon$. After training, generating new data samples can be accomplished by first initializing $\mathbf{v}_T$ with a standard normal distribution, $\mathcal{N}(\mathbf{0}, \mathbf{I})$, and then drawing samples according to $q_\theta(\mathbf{v}_{t-1} \mid \mathbf{v}_t)$ iteratively.

## 3   METHOD

We use a two-stage object detector (Ren et al., 2015) and adopt a main baseline method (Du et al., 2022b) to demonstrate the proposed method and focus on exploring the latent feature space derived from the penultimate layer as depicted in Figure 2. Specifically, we first extract latent space features and utilize them to train a diffusion model to fit the ID data distribution within the compact latent space (see Section 3.1). Next, given the generated samples from the diffusion model, we design a KNN-based density-based filter to obtain low-density samples that can be easily confused with high-density ID samples (see Section 3.2). Finally, the generated OOD samples are used to train our

OOD detector, enabling it to learn a discriminative decision boundary between ID and OOD data (see Section 3.3).

Initially, we commence with the training of the baseline detection model, which consists of the front-end backbone network $f_{\text{fro}}$, the classification head $f_{\text{cls}}$, and the regression head $f_{\text{reg}}$ (see Figure 2), using dataset $\mathcal{D}_{\text{close}}$ containing only ID samples. For a proposal from an ID sample, the front-end network, parameterized by $\theta_{\text{fro}}$, produces feature $\mathbf{v}^{\text{id}} \in \mathbb{R}^{D_{\text{rep}}}$ as follows:

$$\mathbf{v}^{\text{id}} = f_{\text{fro}}(\mathbf{x}^{\text{id}}; \mathbf{b}^{\text{id}}, \theta_{\text{fro}}), \tag{1}$$

which serves as the input to both the classification $f_{\text{cls}}$ and regression head $f_{\text{reg}}$ for classification and localization, respectively. Given $\mathbf{v}^{\text{id}}$, the classification head $f_{\text{cls}}$ and the regression head $f_{\text{reg}}$ produces the output of categorical logits $\mathbf{z}^{\text{id}}$ and bounding box coordinate $\mathbf{b}^{\text{id}}$ as follows:

$$\mathbf{z}^{\text{id}} = f_{\text{cls}}(\mathbf{v}^{\text{id}}; \theta_{\text{cls}}) \quad \text{and} \quad \mathbf{b}^{\text{id}} = f_{\text{reg}}(\mathbf{v}^{\text{id}}; \theta_{\text{reg}}), \tag{2}$$

where $f_{\text{cls}}$ and $f_{\text{reg}}$ are single-layer neural networks parameterized by $\theta_{\text{cls}}$ and $\theta_{\text{reg}}$, respectively.

To promote the compactness of features generated by the front-end network, we employ a prototype-based representation learning loss $\mathbf{L}_{\text{rep}}$ that pulls ID feature representations towards their corresponding class-specific prototypes, thereby making the feature space of each class more compact. More details are shown in the supplementary material. We optimize the front-end network ($\theta_{\text{fro}}$), the classification head ($\theta_{\text{cls}}$), and the regression head ($\theta_{\text{reg}}$) using a combination of the baseline model loss (Du et al., 2022b), and the aforementioned representation learning loss $\mathbf{L}_{\text{rep}}$. The optimized parameters are denoted as $\theta_{\text{fro}}^*$, $\theta_{\text{cls}}^*$ and $\theta_{\text{reg}}^*$ respectively.

## 3.1 DATA DISTRIBUTION MODELING WITH DIFFUSION MODEL

Given the features $\mathbf{v}^{\text{id}}$ obtained from the optimized front-end network, we construct a dataset using proposals from ID samples and their features. Our objective is to utilize the diffusion model to represent the accurate ID data distribution within the latent feature space.

For simplicity, we omit the superscript and refer to $\mathbf{v}^{\text{id}}$ as $\mathbf{v}_0$, denoting it as the starting point of the forward path in the diffusion's training process. Following the standard training procedure of the diffusion model (see Section 2), we perturb a clean data sample $\mathbf{v}_0$ into $\mathbf{v}_t$ and train a denoising network (Ho et al., 2020) to eliminate the noise by minimizing the following loss function:

$$\mathbf{L}_{\text{simple}} = \mathbb{E}_{t, \mathbf{v}_0, \epsilon} \left[ \| \epsilon - \epsilon_\theta (\mathbf{v}_t, t) \|^2 \right], \quad \epsilon \sim \mathcal{N}(\mathbf{0}, \mathbf{I}), \tag{3}$$

where $t$ represents the time step, and $\mathcal{N}(\mathbf{0}, \mathbf{I})$ denotes the unit Gaussian distribution. Once trained, new data can be sampled by initializing $\mathbf{v}_T \sim \mathcal{N}(0, \mathrm{I})$ and then gradually denoising it. The generated features fit the complex distribution of ID data in latent space well and are denoted as $\mathbf{v}^{\text{gen}}$, as shown in Figure 2.

## 3.2 SAMPLE OOD FEATURES

Out-of-distribution (OOD) samples are generally regarded as data points with low probability density under the training set distribution (i.e., ID data distribution) (Bishop, 1994). However, for the diffusion model trained on ID data, the generated data samples $\mathbf{v}^{\text{gen}}$ typically reside in high-density regions (Sehwag et al., 2022). Therefore, We design a filter based on K-Nearest Neighbors (KNN) to select low-density samples at the boundary of the ID distribution as OOD samples.

To estimate the density under ID data distribution, we calculate the distance of each generated sample relative to its $k^*$-th nearest ID sample, where a smaller distance indicates high density (Sun et al., 2022). Consequently, generated data samples with small distances are considered ID samples and are subsequently filtered as follows:

$$\mathbf{v}^{\text{ood}} = \left\{ \mathbf{v}^{\text{gen}} | r_{k^*}(\mathbf{v}^{\text{gen}}; \mathbf{v}^{\text{id}}) \geq \lambda \right\}, \tag{4}$$

where $r_{k^*}(\mathbf{v}^{\text{gen}}; \mathbf{v}^{\text{id}}) = \| \mathbf{v}^{\text{gen}} - \mathbf{v}^{\text{id}}_{(k^*)} \|_2$ represents the Euclidean distance from the generated data to its $k^*$-th nearest ID sample. This filter can effectively exclude generated samples exhibiting high density relative to ID samples without the need for any parametric distribution assumptions.

Furthermore, we also exclude samples that are too far away from the ID data, as they often represent meaningless noise or are easily distinguished from high-density ID samples. Such samples provide little benefit or may even harm the optimization of the decision boundary. The remaining generated OOD samples (as shown in Figure 2) serve as representative data in low-density regions that can also be easily confused with ID data. These samples are then utilized to fine-tune the OOD detector, thereby enhancing its discriminative capabilities for the OOD objects.

## 3.3 OOD DETECTION HEAD

For OOD detection, we follow (Du et al., 2022b) and use the negative energy score of the classifier logits to be the OOD score, where a lower value indicates a higher probability of being an OOD sample. In this step, we augment the original network in (Du et al., 2022b) with a dedicated OOD detection head $f_{\text{ood}}$ which is a single layer LP with parameters $\theta_{\text{ood}}$, to output $K$ classification logits and compute the OOD score. Specifically, for an input $\mathbf{v}$ in the latent feature space, the output logit for the $k$-th category from $f_{\text{ood}}$ can be expressed as:

$$\mathbf{z}_k = f_{\text{ood}}^{(k)}(\mathbf{v}; \theta_{\text{ood}}^{(k)}), \quad k = 1, 2, ..., K. \tag{5}$$

The negative energy score of $\{\mathbf{z}_k | k = 1, 2, ..., K\}$ is the OOD detection score, which is expressed as:

$$-E(\mathbf{v}; \theta_{\text{ood}}) = T \cdot \log \sum_k^K \exp^{\mathbf{z}_k / T} \quad \propto \quad \log p(\mathbf{v}; \theta_{\text{fro}}^*). \tag{6}$$

**Training.** For training, we use the ID data samples $\{\mathbf{v}^{\text{id}}\}$ and generated OOD samples $\{\mathbf{v}^{\text{ood}}\}$ to finetune $f_{\text{ood}}$. Our goal is to encourage it to output a high negative energy score for ID samples and a low negative energy score for OOD samples to obtain parameters $\theta_{\text{ood}}^*$ as:

$$\theta_{\text{ood}}^* = \arg\min_{\theta_{\text{ood}}} \left[ \mathbb{E}_{\mathbf{v} \sim \mathbf{v}^{\text{id}}} \left[ -\log \frac{1}{1 + \exp^{-\phi(E(\mathbf{v}; \theta_{\text{ood}}))}} \right] + \mathbb{E}_{\mathbf{v} \sim \mathbf{v}^{\text{ood}}} \left[ -\log \frac{\exp^{-\phi(E(\mathbf{v}; \theta_{\text{ood}}))}}{1 + \exp^{-\phi(E(\mathbf{v}; \theta_{\text{ood}}))}} \right] \right]. \tag{7}$$

**Inference.** The training aims to encourage $\theta_{\text{ood}}^*$ to satisfy the following condition:

$$-E_{\mathbf{x} \notin \mathcal{D}_{\text{close}}}(\mathbf{v}; \mathbf{x}, \mathbf{b}, \theta_{\text{ood}}^*) < -E_{\mathbf{x} \in \mathcal{D}_{\text{close}}}(\mathbf{v}; \mathbf{x}, \mathbf{b}, \theta_{\text{ood}}^*). \tag{8}$$

Hence, for inference, we employ the negative energy score from the optimized OOD detection head to determine whether an evaluating object is an OOD or ID instance. If the negative energy score of an object exceeds a certain threshold, the sample is classified as an ID sample. A category label from the classification head is then assigned to the detected ID object. Otherwise, it is assigned as an OOD object.

## 4 EXPERIMENT

**Datasets** We use two datasets as the ID training data: **PASCAL VOC** (Everingham et al., 2010) and **Berkeley DeepDrive (BDD-100K)** (Yu et al., 2020), comprising 20 and 10 ID categories, respectively. For the models trained on these two training sets, we evaluate them on two OOD datasets: **MS-COCO** (Lin et al., 2014) and **OpenImages** (Kuznetsova et al., 2020) (validation set). Following the baseline setting (Du et al., 2022b), categories from the OOD datasets that overlapped with the ID datasets are removed to guarantee the absence of ID categories.

**Metrics** In this paper, we primarily focus on reporting the **FPR95↓**, which represents the false positive rate of OOD samples when the true positive rate of ID samples is at 95%, which is widely used to assess the OOD detection performance. Additionally, we present the area under the receiver operating characteristic curve (**AUROC↑**), and the area under the precision-recall curve (**AUPR↑**), which are widely utilized to evaluate binary classification problems. Furthermore, we report the mean average precision (**mAP↑**) on ID data, a general metric used to estimate the ID object detection capability of one detection model.

**Implementation Details** Unless otherwise specified, we follow the setting of baseline method (Du et al., 2022b) to train the object detector with ResNet-50 (He et al., 2016) as the backbone firstly, while the representation learning loss weight is set as 0.01. At the fine-tuning phase, we apply a

Table 1: Comparison with varied ID (PASCAL VOC (Everingham et al., 2010), BDD-100K (Yu et al., 2020)) and OOD (MS-COCO (Lin et al., 2014), OpenImages (Kuznetsova et al., 2020)) datasets. Our method significantly outperforms other methods on different metrics and achieves state-of-the-art performance. (* indicates methods re-implemented by ourselves)

| | Method | OOD: MS-COCO | | | OOD: OpenImages | | | mAP ↑ |
|---|---|---|---|---|---|---|---|---|
| | | FPR95 ↓ | AUROC ↑ | AUPR ↑ | FPR95 ↓ | AUROC ↑ | AUPR ↑ | |
| **ID: PASCAL-VOC** | MSP (Hendrycks & Gimpel, 2017) | 70.99 | 83.45 | - | 73.13 | 81.91 | - | 48.7 |
| | ODIN (Liang et al., 2018) | 59.82 | 82.20 | - | 63.14 | 82.59 | - | 48.7 |
| | Mahalanobis (Lee et al., 2018b) | 96.46 | 59.25 | - | 96.27 | 57.42 | - | 48.7 |
| | Energy score (Liu et al., 2020) | 56.89 | 83.69 | - | 58.69 | 82.98 | - | 48.7 |
| | Gram matrices (Sastry & Oore, 2020a) | 62.75 | 79.88 | - | 67.42 | 77.62 | - | 48.7 |
| | Generalized ODIN (Hsu et al., 2020) | 58.57 | 83.12 | - | 70.28 | 79.23 | - | 48.1 |
| | CSI (Tack et al., 2020) | 59.91 | 81.83 | - | 57.41 | 82.95 | - | 48.1 |
| | GAN-synthesis (Lee et al., 2018a) | 60.93 | 83.67 | - | 59.97 | 82.67 | - | 48.5 |
| | VOS (Du et al., 2022b) | 47.53 | 88.70 | 98.98 | 51.33 | 85.23 | 97.40 | 48.9 |
| | SIREN-KNN* (Du et al., 2022a) | 49.09 | 88.85 | 98.98 | 53.04 | 87.61 | 98.04 | 48.4 |
| | *Baseline* | 48.19 | 88.91 | 98.98 | 54.77 | 84.48 | 97.40 | 48.9 |
| | *Ours* | 45.59 | 89.30 | 99.01 | 50.19 | 87.07 | 97.95 | 49.1 |
| **ID: BDD-100K** | MSP (Hendrycks & Gimpel, 2017) | 80.94 | 75.87 | - | 79.04 | 77.38 | - | 31.2 |
| | ODIN (Liang et al., 2018) | 62.85 | 74.44 | - | 58.92 | 76.61 | - | 31.2 |
| | Mahalanobis (Lee et al., 2018b) | 57.66 | 84.92 | - | 60.16 | 86.88 | - | 31.2 |
| | Energy score (Liu et al., 2020) | 60.06 | 77.48 | - | 54.79 | 79.60 | - | 31.2 |
| | Gram matrices (Sastry & Oore, 2020a) | 60.93 | 74.93 | - | 77.55 | 59.38 | - | 31.2 |
| | Generalized ODIN (Hsu et al., 2020) | 57.27 | 85.22 | - | 50.17 | 87.18 | - | 31.8 |
| | CSI (Tack et al., 2020) | 47.10 | 84.09 | - | 37.06 | 87.99 | - | 30.6 |
| | GAN-synthesis (Lee et al., 2018a) | 57.03 | 78.82 | - | 50.61 | 81.25 | - | 31.4 |
| | VOS (Du et al., 2022b) | 44.27 | 86.87 | 99.70 | 35.54 | 88.52 | 99.84 | 31.3 |
| | SIREN-KNN* (Du et al., 2022a) | 45.83 | 90.28 | 99.85 | 39.93 | 90.45 | 99.89 | 31.3 |
| | *Baseline* | 51.91 | 83.72 | 99.70 | 42.72 | 86.41 | 99.84 | 31.2 |
| | *Ours* | 38.90 | 90.31 | 99.79 | 30.37 | 91.86 | 99.90 | 31.0 |

learning rate of 1e-4 for training and execute 100 iterations on the generated OOD data. The main training is conducted on GeForce RTX 3090 GPUs. Meanwhile, the fine-tuning is implemented either on a single GeForce RTX 3090 GPU or an Apple M2 Pro Chip.

In the following, we present our main results and compare them with baseline methods in Section 4.1. Then, we carry out ablation studies and conduct comprehensive analyses of various design elements, such as the diffusion model, filtering scheme, representation learning loss, and different feature spaces adopted for data generation in Section 4.2 and 4.3.

## 4.1 MAIN RESULTS

We evaluate the performance of the proposed approach on different challenging benchmarks. As is depicted in Table 1, compared with the baseline method, our method shows a significant improvement in FPR95 while maintaining comparable ID object detection capability. Notably, around 13% gain in FPR95 is achieved when the ID dataset is BDD compared to the baseline method. Meanwhile, the proposed method consistently outperforms other approaches. For a fair comparison, following the benchmark of OpenOOD (Yang et al., 2022), all the methods shown in Table 1 only use ID object detection data without using extra OOD data.

We also provide a qualitative comparison between our proposed method and the baseline model, using COCO as the OOD dataset. Our results, shown in Figure 3, demonstrate that our method successfully identifies OOD samples that are prone to be confused. The baseline model often misclassifies unseen objects as ID objects with high confidence, *e.g.*, misclassifying zebras as motorbikes with high negative energy and confidence score due to their structural similarity. In contrast, our model assigns these OOD objects significantly lower negative energy scores, thus accurately identifying them as OOD samples. Similar conclusions can be drawn for other instances.

To further demonstrate the generalizability of our method, we also extend the evaluation with multiple backbones. Regarding the results using RegNetX-4.0GF (Radosavovic et al., 2020), in comparison to the baseline method, which exhibits FPR95 / AUROC / AUPR of 47.27 / 89.35 / 99.10 (%) and 47.71 / 88.17 / 98.27 (%) on two OOD datasets (ID: PASCAL-VOC), our method achieves superior results

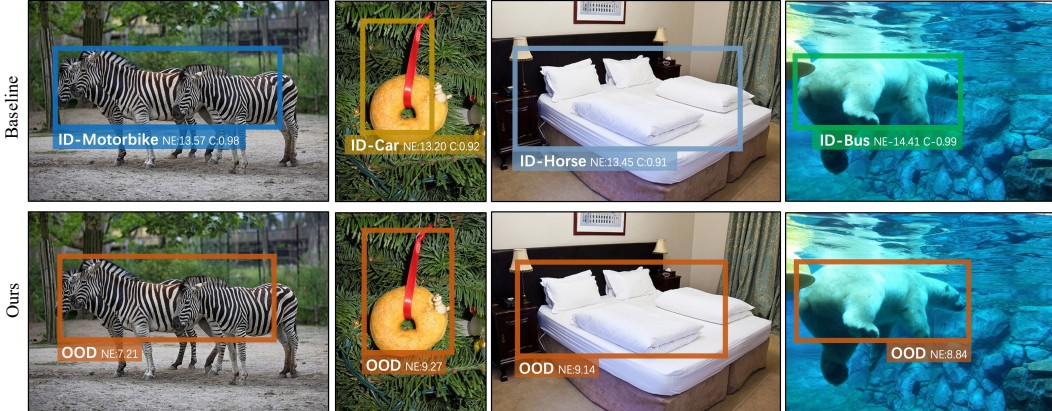

Figure 3: Qualitative results on the MS-COCO validation dataset (ID dataset: PASCAL-VOC). The baseline method classifies objects from unseen categories as ID samples with high negative energy scores and assigns them as wrong categories with high confidence. In contrast, our method assigns these objects lower negative energy scores and thus classifies them as OOD samples. (NE indicates negative energy score; C indicates classification confidence score.)

Table 2: Comparison on representation generation method (ID dataset: PASCAL-VOC). Starting from the same latent space, we comparatively use different methods to generate OOD samples and fine-tune the OOD detection head. Our method (diffusion + filter) can generate better OOD features than any other method.

| Method | OOD: MS-COCO | | | OOD: OpenImages | | |
|---|---|---|---|---|---|---|
| | FPR95 $\downarrow$ | AUROC $\uparrow$ | AUPR $\uparrow$ | FPR95 $\downarrow$ | AUROC $\uparrow$ | AUPR $\uparrow$ |
| GMM$_{+negative\ sampling}$ | 58.02 | 85.82 | 98.68 | 62.00 | 82.44 | 96.99 |
| OpenGAN | 46.87 | 89.37 | 99.01 | 51.35 | 87.30 | 98.05 |
| VAE$_{+filter}$ | 46.71 | 89.60 | 99.03 | 48.65 | 88.01 | 98.14 |
| GAN$_{+filter}$ | 46.05 | 88.88 | 99.03 | 49.86 | 87.57 | 98.11 |
| GMM$_{+filter}$ | 45.88 | 89.45 | 99.02 | 48.28 | 87.54 | 98.06 |
| Background | 45.55 | 89.54 | 99.03 | 48.79 | 87.95 | 98.13 |
| *Ours* (diffusion + filter) | 44.98 | 89.60 | 99.03 | 47.85 | 87.99 | 98.14 |

of 44.87 / 90.01 / 99.13 (%) and 44.51 / 89.50 / 98.47 (%), respectively. In addition, we also evaluate the performance of the proposed method on the OOD image classification task with DenseNet (Huang et al., 2017) as the backbone, and our results still stably surpass the baseline. Results are provided in the supplementary material.

## 4.2 ABLATION STUDIES

**Effectiveness of OOD Feature Generation**  We carry out a comprehensive comparison with other methods of generating OOD samples (see Table 2) to demonstrate the efficacy of utilizing diffusion models, while also investigating their combined effects once combined with the filtering scheme. Though all methods work in the same feature space, samples generated with GMM + negative sampling may not be critical to training the OOD detector, which achieves very poor results. The OOD features generated by OpenGAN (Kong & Ramanan, 2021) are better than the previous method but are still worse than our method. Furthermore, combining our designed filtering scheme as depicted in equation 4, VAE (Kingma & Welling, 2013), GAN (Goodfellow et al., 2014), and GMM can generate OOD features that improve performance but still lag behind ours. Although the filter we designed can very well help select OOD samples that are close to the ID data, GMM makes wrong assumptions about the complex ID data distribution, resulting in the generated features being unable to describe the ID distribution space well. In comparison, our method utilizes a diffusion model to fit accurate ID distribution and exhibits superior performance. This fully proves the effectiveness of our

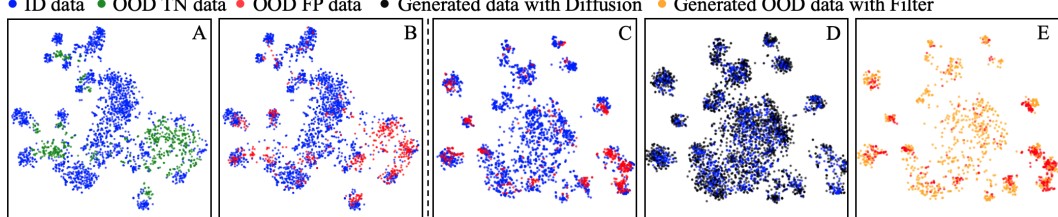

Figure 4: T-SNE (Van der Maaten & Hinton, 2008) visualization of the penultimate layer feature space for baseline model (A and B) and our model before fine-tuning (C, D, and E). For the baseline model, we plot ID data, OOD true negative, and false positive data. For our model, we plot ID data, OOD false positive data, generated data with diffusion model, and the selected OOD data with the proposed filter for fine-tuning (ID: PASCAL-VOC, OOD: MS-COCO).

diffusion model. Moreover, OOD features that are extracted from background proposals do not bring as much benefit as the features generated by our method.

**Effectiveness of Proposed Filter** We examine the impact of our proposed filtering scheme during the fine-tuning of the OOD detector head. As illustrated in Table 3, when data are directly sampled from the trained diffusion model without the filtering strategy ( *w.o. filter*) and utilized to fine-tune the model, a decline in the OOD detection capability relative to the baseline (*w.o. f.t.*) is observed. This is because the generated data contains a large number of high-density ID

Table 3: Comparison on filters (FPR95 / AUROC / AUPR are reported on two OOD datasets with PASCAL-VOC as ID dataset).

| Method | MS-COCO | OpenImages |
|---|---|---|
| *w.o. f.t.* | 46.95 / 88.95 / 98.98 | 51.69 / 86.64 / 97.79 |
| *w.o. filter* | 47.71 / 88.65 / 98.94 | 52.03 / 86.05 / 97.73 |
| GMM | 46.93 / 89.24 / 99.00 | 50.96 / 87.01 / 97.97 |
| KNN (*Ours*) | 44.98 / 89.60 / 99.03 | 47.85 / 87.99 / 98.14 |

samples. By using our KNN-based filtering scheme (KNN (*ours*)), the performance has been significantly improved. In addition, we explore the utilization of a parametric model, GMM, to model the ID data distribution and select low-density OOD samples (GMM). Although the performance improves, it still achieves inferior performance compared to our KNN-based filtering scheme, which supports the efficacy of our KNN-based filter.

**Discussion of the Feature Space for Generation** In our method, we encode ID samples into the compact latent space and model the underlying distribution with a diffusion model, as opposed to fitting the distribution in the original image space. This approach brings two advantages: 1) The original image space is relatively redundant, encompassing many subtle details that do not significantly contribute to detection and classification. The latent feature space, on the other hand, is more compact. Moreover, owing to our representation learning loss, this space focuses more on semantic information, which aligns with the interests of the classifier; 2) The dimension of the feature space is relatively lower, thereby reducing the fitting complexity and the difficulty for a generative model to manipulate the given feature. We use the diffusion model to generate samples in RGB object space and use the same filtering strategy to generate OOD samples to fine-tune the OOD detection head. The results show that, from the same start feature space, this method performs worse on FPR95 than using the penultimate layer features on average 0.9%. Furthermore, we have explored generating OOD samples on the feature space before RPN with a dimensionality of ($D_{\mathrm{rep}} = 7 \times 7 \times 256$) (Ren et al., 2015) and fine-tuning the model. However, the fine-tuned model only brought improvements of 0.13% and 0.30% on FPR95 on the two OOD datasets compared to not fine-tuned, respectively.

## 4.3 FEATURE SPACE VISUALIZATION

We use t-SNE (Van der Maaten & Hinton, 2008) to visualize the penultimate layer feature space of the baseline model (A and B) and our model before fine-tuning (C, D, and E). As shown in Figure 4, the baseline model performs better on OOD samples far away from ID data (sub-figure A, green points: identify true OOD as OOD), but performs poorly in areas that are easily confused with ID data (sub-figure B: red points: recognize true OOD as ID). This observation well supports our motivation. When we use representation loss to train the model, "hard cases" near ID data still exist (sub-figure C). However, we use the diffusion model to generate features (black points in sub-figure D), that fit the distribution of ID data well, and then use the proposed filter to obtain OOD features (orange

points in sub-figure E). The generated OOD features well cover the false positive OOD samples, which provides strong support for the effectiveness of our proposed method. More failure cases of the baseline model are shown in the supplementary material.

## 5 RELATED WORK

**OOD Detection for Object Detection**    As for OOD detection in object detection, VOS (Du et al., 2022b) synthesize virtual outliers based on class-conditional Gaussian Distribution in the latent feature space for model training regularization and SIREN (Du et al., 2022a) learns a pre-defined distribution on ID data. However, our work employs a diffusion model for generating out-of-distribution data for model training regularization, without hypothesizing a prior data distribution. Also, highly relevant open-set object detection is generally related to open-world object detection and OOD detection. Pioneering works (ORE (Joseph et al., 2021) and OWDETR (Gupta et al., 2022)) of open-world object detection often use object priors to identify unknown objects. Open-world object tasks can also exploit round-robin learning approach (French, 1999; McCloskey & Cohen, 1989; Wang et al., 2020), out-of-domain generalization (Wang et al., 2021c), and zero-shot object detection (Rahman et al., 2020). Approximating Bayesian methods, such as MC-Dropout (Miller et al., 2018; 2019; Deepshikha et al., 2021; Dhamija et al., 2020; Hall et al., 2020), are also used for OOD detection.

**OOD Detection for Image Classification**    The first paradigm includes the OpenMax score developed by Bendale & Boult (2016) that uses extreme value theory (EVT), while Hendrycks & Gimpel (2017) proposes a simple baseline using maximum softmax probability. Further developments include deep ensemble (Lakshminarayanan et al., 2017), ODIN (Liang et al., 2018), distance-based score (Lee et al., 2018b; Ren et al., 2021; Sastry & Oore, 2020b; Sun et al., 2022), energy-based score (Liu et al., 2020; Wang et al., 2021a), DICE (Sun & Li, 2022) and ReAct score (Sun et al., 2021b). While post hoc methods often consider the OOD scoring function alone, our framework takes both representation learning (at training) and OOD detection (at testing) into account. Regularization-based methods like those by Tack et al. (2020) and Sun et al. (2020) focus on modulating model output (Hendrycks et al., 2018; Liu et al., 2020; Du et al., 2022b) and shaping latent representations. The assumption made by SIREN (Du et al., 2022a) and VOS (Du et al., 2022b) - that the in-distribution training data adheres to an explicit distribution in the latent feature space - may not always hold. Consequently, this reliance can result in sub-optimal models. Our method proposes a more reliable solution by factoring in both representation learning and OOD detection.

**Diffusion Models**    Diffusion models (Ho et al., 2020; Song & Ermon, 2020) have recently attracted significant attention in the community due to their powerful content generation capabilities, including the generation of 2D images (Nichol & Dhariwal, 2021; Ramesh et al., 2022; Nichol et al., 2022; Saharia et al., 2022; Rombach et al., 2022; Dhariwal & Nichol, 2021). Unlike discriminative models commonly used in perception tasks, diffusion models, similar to GANs (Goodfellow et al., 2014), VAEs (Kingma & Welling, 2013), and Normalizing Flows (Papamakarios et al., 2021), are generative models, whose essence is to model the probability distribution of the data itself, enabling it to be controlled. Therefore, beyond direct applications in the content generation domain, diffusion models have also been gradually explored and utilized in perception tasks (He et al., 2022; Azizi et al., 2023; Xu et al., 2023; Baranchuk et al., 2021) in recent years. In our work, we leverage diffusion models to faithfully model the distribution of our feature space, further designing it to generate Out-of-Distribution (OOD) features of interest. To the best of our knowledge, ours is the first work that uses diffusion models to assist in OOD detection.

## 6 CONCLUSION

In this paper, we have presented OpenDiffusion, which leverages the diffusion model to generate data samples and incorporates a filtering mechanism to produce meaningful out-of-distribution (OOD) data for optimizing the OOD detector. The diffusion model can faithfully represent the in-distribution (ID) data distribution, overcoming the limitations inherent in conventional distribution assumptions. Moreover, our KNN-based filter is designed to select OOD samples that not only have low probability density but also contribute positively to training the OOD detector. Extensive experiments on multiple OOD benchmarks demonstrate the effectiveness of our proposed approach. We hope our investigation will inspire further research into exploring generative models for recognition problems.

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
