# SUPPLEMENTARY MATERIAL

## 1 DIFFUSION MODEL

**Forward process.** The Denoising Diffusion Probabilistic Model (DDPM) (Nichol & Dhariwal, 2021) aims to model the training data distribution. Given a sample from the training data, e.g., $v_0 \sim q(\mathbf{v}_0)$, the forward diffusion process $q\left(\mathbf{v}_{1:T} \mid \mathbf{v}_0\right) = \prod_{t=1}^{T} q\left(\mathbf{v}_t \mid \mathbf{v}_{t-1}\right)$ incrementally distorts the data using Gaussian kernels $q\left(\mathbf{v}_t \mid \mathbf{v}_{t-1}\right) := \mathcal{N}\left(\sqrt{1 - \beta_t}\mathbf{v}_{t-1}, \beta_t\mathbf{I}\right)$, thereby generating progressively distorted hidden variables $\mathbf{v}_1, \mathbf{v}_2, ..., \mathbf{v}_T$. Remarkably, $v_t$ can be sampled from the following probability density:

$$q\left(\mathbf{v}_t \mid \mathbf{v}_0\right) = \mathcal{N}\left(\mathbf{v}_t; \sqrt{\bar{\alpha}_t}\mathbf{v}_0, \left(1 - \bar{\alpha}_t\right)\mathbf{I}\right), \tag{1}$$

where we define $\alpha_t := 1 - \beta_t$ and $\bar{\alpha}_t := \prod_{s=1}^{t} \alpha_s$. Generally, the forward trajectory variances $\beta_t$ are held constant and progressively escalated from $\beta_1 = 10^{-4}$ to $\beta_T = 0.02$. Additionally, $T$ should be sufficiently large (e.g., 1000) to ascertain $q\left(\mathbf{v}_T \mid \mathbf{v}_0\right) \approx \mathcal{N}(0, \mathbf{I})$. The diffusion model's objective is to characterize the joint distribution $q\left(\mathbf{v}_{0:T}\right)$, which includes a tractable sampling path for the marginal distribution $q\left(\mathbf{v}_0\right)$.

**Reverse process.** The reverse diffusion process aims to recover $\mathbf{v}_0$ from a random sample $\mathbf{v}_T$:

$$p_\theta\left(\mathbf{v}_{0:T}\right) := p\left(\mathbf{v}_T\right)\prod_{t=1}^{T} p_\theta\left(\mathbf{v}_{t-1} \mid \mathbf{v}_t\right),$$
$$p_\theta\left(\mathbf{v}_{t-1} \mid \mathbf{v}_t\right) := \mathcal{N}\left(\mathbf{v}_{t-1}; \boldsymbol{\mu}_\theta\left(\mathbf{v}_t, t\right), \boldsymbol{\Sigma}_\theta\left(\mathbf{v}_t, t\right)\right), \tag{2}$$

where a deep neural network with parameters $\theta$ is used to predict the mean $\boldsymbol{\mu}_\theta$ and covariance matrix $\boldsymbol{\Sigma}_\theta$ of the probability density function. Learning is achieved by optimizing a variational upper bound of the negative log-likelihood of the training data:

$$\mathrm{E}_{q(\mathbf{v}_0)}\left[-\log p_\theta\left(\mathbf{v}_0\right)\right] \leq \mathrm{E}_{q(\mathbf{v}_{0:T})}\left[-\log \frac{p_\theta\left(\mathbf{v}_{0:T}\right)}{q\left(\mathbf{v}_{1:T} \mid \mathbf{v}_0\right)}\right] =: L. \tag{3}$$

This loss term $L$ can be represented as:

$$L = \mathbb{E}_q[\underbrace{D_{\mathrm{KL}}\left(q\left(\mathbf{v}_T \mid \mathbf{v}_0\right)\|p\left(\mathbf{v}_T\right)\right)}_{L_T}$$
$$+ \sum_{t>1}\underbrace{D_{\mathrm{KL}}\left(q\left(\mathbf{v}_{t-1} \mid \mathbf{v}_t, \mathbf{v}_0\right)\|p_\theta\left(\mathbf{v}_{t-1} \mid \mathbf{v}_t\right)\right)}_{L_{t-1}} \underbrace{-\log p_\theta\left(\mathbf{v}_0 \mid \mathbf{v}_1\right)}_{L_0}]. \tag{4}$$

The optimization terms $L_{t-1}(t > 1)$ can be computed analytically since the two terms being compared in the KL divergence are both Gaussians, i.e.,

$$q\left(\mathbf{v}_{t-1} \mid \mathbf{v}_t, \mathbf{v}_0\right) = \mathcal{N}\left(\mathbf{v}_{t-1}; \tilde{\boldsymbol{\mu}}_t\left(\mathbf{v}_t, \mathbf{v}_0\right), \tilde{\beta}_t\mathbf{I}\right),$$
$$p_\theta\left(\mathbf{v}_{t-1} \mid \mathbf{v}_t\right) := \mathcal{N}\left(\mathbf{v}_{t-1}; \boldsymbol{\mu}_\theta\left(\mathbf{v}_t, t\right), \boldsymbol{\Sigma}_\theta\left(\mathbf{v}_t, t\right)\right), \tag{5}$$

where $\tilde{\boldsymbol{\mu}}_t\left(\mathbf{v}_t, \mathbf{v}_0\right) := \frac{\sqrt{\bar{\alpha}_{t-1}}\beta_t}{1-\bar{\alpha}_t}\mathbf{v}_0 + \frac{\sqrt{\alpha_t}(1-\bar{\alpha}_{t-1})}{1-\bar{\alpha}_t}\mathbf{v}_t$ and $\tilde{\beta}_t := \frac{1-\bar{\alpha}_{t-1}}{1-\bar{\alpha}_t}\beta_t$. DDPM (Ho et al., 2020) fix $\boldsymbol{\Sigma}_\theta\left(\mathbf{v}_t, t\right) = \sigma_t^2\mathbf{I}$ during training, where $\sigma_t^2$ is set to be $\beta_t$ or $\tilde{\beta}_t$. In our experiments, we set $\sigma_t^2 = \beta_t$.

Table 1: The OOD detection results on image classification task, using CIFAR-10 as ID dataset and DenseNet (Huang et al., 2017) as model architecture.

| OOD Data | Baseline | | | Ours | | |
|---|---|---|---|---|---|---|
| | FPR95 ↓ | AUROC ↑ | AUPR ↑ | FPR95 ↓ | AUROC ↑ | AUPR ↑ |
| Texture | 50.03 | 89.03 | 96.97 | 49.48 | 89.14 | 97.19 |
| SVHN | 41.18 | 93.94 | 98.80 | 43.49 | 93.19 | 98.63 |
| Places365 | 46.11 | 89.69 | 97.47 | 43.24 | 90.56 | 97.71 |
| LSUN-C | 4.93 | 98.85 | 99.79 | 4.93 | 98.91 | 99.78 |
| LSUN-Resize | 10.44 | 97.97 | 99.61 | 5.72 | 98.56 | 99.72 |
| iSUN | 12.58 | 97.69 | 99.55 | 6.14 | 98.50 | 99.71 |
| *Average* | 27.55 | 94.55 | 98.70 | 25.50 | 94.81 | 98.79 |

**Loss function.** In practice, the model can be trained to predict $\boldsymbol{\mu}_t$, $\mathbf{v}_0$ or $\boldsymbol{\epsilon}$ by changing the specific way of parameterization, as illustrated in (Ho et al., 2020). When predicting $\boldsymbol{\epsilon}$ (Ho et al., 2020), the final training term is simplified (Ho et al., 2020) as follows:

$$L_{\text{simple}}(\theta) := \mathbb{E}_{t,\mathbf{v}_0,\epsilon}\left[\left\|\boldsymbol{\epsilon} - \boldsymbol{\epsilon}_\theta\left(\sqrt{\bar{\alpha}_t}\mathbf{v}_0 + \sqrt{1-\bar{\alpha}_t}\boldsymbol{\epsilon}, t\right)\right\|_2^2\right],\tag{6}$$

where $\epsilon \sim \mathcal{N}(0, \mathbf{I})$ and $t$ is uniformly sampled between 1 and $T$. In our experiments, we predict $\mathbf{v}_0$ since we find it results in more stable training. The loss function is modified as:

$$L_{\text{v}_0}(\theta) := \mathbb{E}_{t,\mathbf{v}_0,\epsilon}\left[\left\|\mathbf{v}_0 - G_\theta\left(\sqrt{\bar{\alpha}_t}\mathbf{v}_0 + \sqrt{1-\bar{\alpha}_t}\boldsymbol{\epsilon}, t\right)\right\|_2^2\right],\tag{7}$$

where $G_\theta$ is the network to be optimized.

## 2 LEARNING A COMPACT LATENT SPACE

For learning a compact latent space duriing training, we introduce a prototype-based representation learning loss to pull ID feature representations towards their corresponding class-specific prototypes. Following the notations in Method Section of the manuscript, for category $k$, we obtain its prototype $\mathbf{p}_k$ by using a memory bank to store features of samples belonging to this class and calculating their average as the prototype:

$$\mathbf{p}_k = \frac{1}{N_k}\sum_{i:\text{y}_i^{\text{id}}=k}\mathbf{v}_i^{\text{id}},\tag{8}$$

where $N_k$ denotes the number of proposals in class $k$. The memory bank maintains a first-in-first-out queue of size $N_k$ for each class (Du et al., 2022), which is consistent across all categories. During training, the queue for each category accumulates $N_k$ proposal features and continuously updates them with the most recent samples. The average of the representations stored in the queue serves as the dynamically updated prototype for each class.

We subsequently impose constraints on features of ID samples and their corresponding class-specific prototypes by considering their Euclidean distances, resulting in the following representation loss:

$$\mathbf{L}_{\text{rep}} = \frac{1}{N}\sum_i\left[\text{y}_i^{\text{id}} = k\right]\left\|\mathbf{v}_i^{\text{id}} - \mathbf{p}_k\right\|_2^2,\tag{9}$$

where $[\cdot]$ is the Iverson bracket and $N$ is the number of all proposals. The representation learning loss promotes the generation of a compact latent feature space, which is beneficial for discriminating ID samples. Additionally, the compact latent space enhances the learning of our generative model and strengthens the identification of OOD samples.

## 3 MORE EXPERIMENTS

### 3.1 RESULTS ON OOD IMAGE CLASSIFICATION

We employ DenseNet (Huang et al., 2017) as the backbone to evaluate the performance of the proposed method on OOD image classification task to demonstrate the generalizability of our approach. The

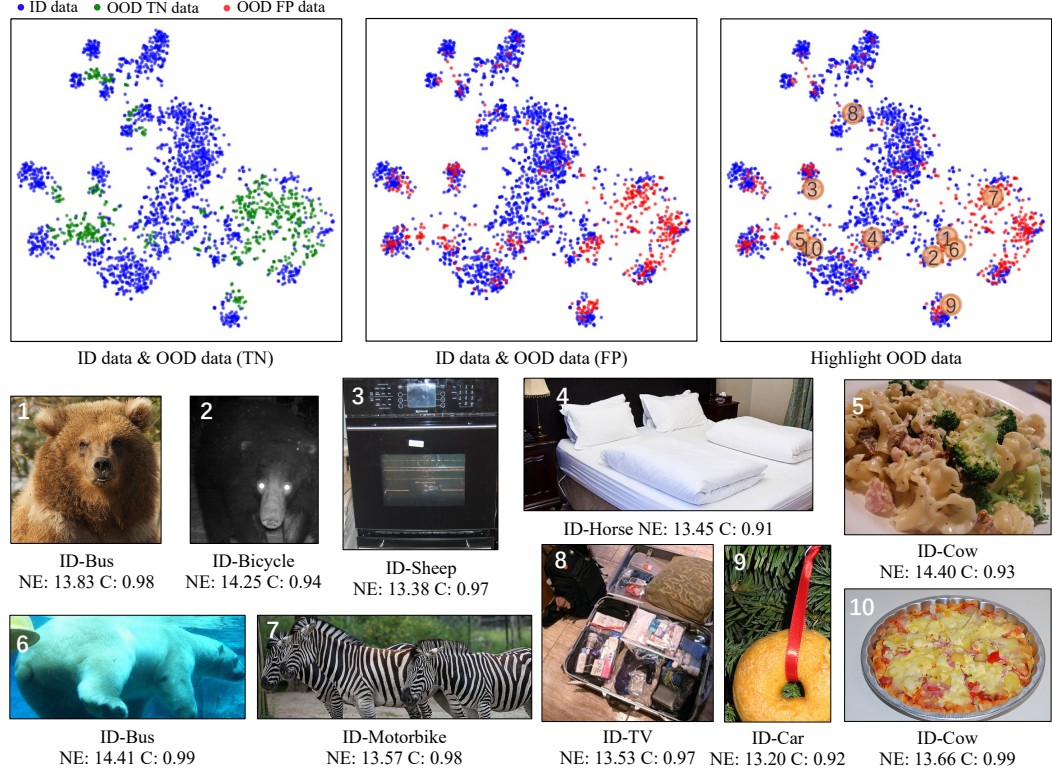

Figure 1: Detailed results on the MS-COCO validation dataset (ID dataset: PASCAL-VOC) of the baseline method. For the visualization on the latent space (top row) with t-SNE (Van der Maaten & Hinton, 2008), the ID samples are shown as blue points, while true negative (TN) and false positive (FP) OOD samples are shown as green and red points, respectively. Some FP OOD proposals are shown (bottom row) with numbers and shown as the relative orange circles in the latent space (top right). (NE indicates negative energy score; C indicates classification confidence score.)

results is shown in Table 1. Our method performs better than baseline methods on OOD image classification task.

## 3.2 FAILURE CASES OF BASELINE MODEL

As an extension of the main experiments and latent space visualization in the manuscript, we first visualize results of the baseline method as shown in Figure 1. When training is completed, we illustrate the true negative (TN) cases, *an OOD sample is classified correctly as OOD*, and false positive (FP) cases, *an OOD sample is misclassified as ID sample*, in the latent space, respectively. It can be observed that compared with the TN OOD samples far away from the ID data distribution, FP OOD samples are closer to the ID data distribution within the latent space. This indicates that the samples most likely to mislead the OOD detectors are those situated near the ID data manifold. These samples hold greater significance for improving the OOD detector, thereby providing further validation for our proposed filtering mechanism's design.

Besides, we present a visualization of 10 proposals related to FP OOD samples in Figure 1, which are classified as ID samples (Du et al., 2022) by the baseline model with high negative energy scores. These samples are further misclassified into incorrect ID categories with high confidence scores. In contrast, our proposed model assigns lower negative energy scores and correctly identifies these instances as OOD samples.

## 3.3 IMPLEMENTATION AND DATASETS DETAILS

For the details of implementation on our method, baseline method, and all comparative experiments, we use the base framework of Faster-RCNN (Ren et al., 2015) with ResNet-50 (He et al., 2016) as

backbone. For the number of negative sampling from the baseline method (Du et al., 2022), we set it as 4 (per-class) to learn the latent space. Our diffusion model $G_\theta$ is derived from the guided diffusion model presented in Dhariwal & Nichol (2021). Considering the target for generation is a vector, we utilize point-wise convolution as basic block and stack seven residual blocks, each containing 1024 channels, whose architecture is identical to that in Dhariwal & Nichol (2021). A dropout ratio of 0.1 is adopted, and the diffusion model is trained for a total of 200 epochs. Besides, the $k$ equals 100 in the KNN-based filter. The 20 classes in Pascal-VOC(Everingham et al., 2010) are all utilized to train the detector. So does the BDD-100K(Yu et al., 2020) dataset that contains 10 categories.

## 4    LIMITATION ANALYSIS

The problem we investigate can classify proposals into $K+1$ categories ($K$ ID categories and 1 OOD category), but it does not further explore what categories the OOD objects belong to. As depicted in Figure 1, although the detector is only trained on the ID data, the distance in the latent space between any two of the three "bears" (1, 2, 6) is very small. Similarly, two visually similar plates of food (5, 10) are very similar in the latent space. This observation demonstrates the potential for discovering the novel categories within OOD samples in the learned space.

For open-world perception tasks, beyond identifying novel objects, it is also necessary to continuously learn the semantic labels of new categories step by step, which is very useful for applications such as autonomous driving and robots manipulation. The proposed approach takes a step forward for the open-world object detection and we will explore it in the future.