# OpenReview forum: "Diffusion-based Data Generation for Out-of-Distribution Object Detection"
_ICLR.cc/2024/Conference — ICLR 2024 Conference Withdrawn Submission_

### Official Review · Reviewer_7pu6 · 2023-10-13

**Soundness:** 2 fair
**Presentation:** 2 fair
**Contribution:** 1 poor
**Rating:** 1
**Confidence:** 5

**Summary:**

This paper aims to address the problem of out-of-distribution object detection. Particularly, this paper proposes a method that leverages the diffusion model to accurately model the ID data distribution and generate OOD data samples. Concretely, the proposed method involves three steps: training a diffusion model to fit the data distribution within the compact latent space, designing a KNN-based density-based filter to obtain low-density samples, and using these samples to train the OOD detector, which is very complex. In the experiments, the proposed method is evaluated on two datasets. And this paper does not outperform the compared method significantly. And the authors do not compare with state-of-the-art methods.

**Strengths:**

Leveraging diffusion models to generate OOD data is meaningful.

**Weaknesses:**

1. The motivation of this paper is not clear. In Introduction Section, the authors directly indicate that they propose OpenDiffusion to leverage the diffusion model to fit data distributions and generate data samples. However, I am not clear why the proposed method could address the analyzed problem. The authors should give more interpretations. Meanwhile, why the authors do not use other generative methods, e.g., VAE and GANs?

2. For OOD task, since diffusion method has impressive generative performance, it is an commonly used mechanism to generate data. Thus, the proposed idea is not novel. Meanwhile, this paper does not sufficiently analyze the corresponding advantages of the proposed diffusion method. Finally, the proposed method mainly includes there steps: training a diffusion model to fit the data distribution within the compact latent space, designing a KNN-based density-based filter to obtain low-density samples, and using these samples to train the OOD detector, which is very complex.

3. This paper does not give any theoretical contributions. In the experiments, for PASCAL VOC dataset, the proposed method does not outperform VOS significantly. The authors should give more interpretations. Meanwhile, the proposed method only compare VOS and SIREN-KNN. And it does not compare state-of-the-art methods, e.g., the methods from CVPR 2023 and ICCV 2023. To the best of my knowledge, the performance of this paper is significantly weaker than state-of-the-art methods.

4. The proposed method is very complex. The proposed method involves multiple operations. The authors should give more ablation experiments. The visualization in Figure 4 is not clear. The authors should give more interpretations.

**Questions:**

This paper does not compare state-of-the-art methods. And this paper does not give any inspired ideas.

---

### Official Review · Reviewer_LV47 · 2023-10-14

**Soundness:** 1 poor
**Presentation:** 3 good
**Contribution:** 1 poor
**Rating:** 3
**Confidence:** 4

**Summary:**

This paper proposes to improve out-of-distribution (OOD) detection performance by generating pseudo-outlier data. Such data is collected by training the diffusion model in in-distribution data and sampling low-density generated data. Then, the OOD detection head is fine-tuned to discriminate such outlier data.

**Strengths:**

(1) I found it easy to understand the paper.\
(2) The idea is simple and easy to implement.

**Weaknesses:**

(1) The paper asserts that they are the first work to apply the diffusion model to generate outlier data. See [1] for the example of previous work. I think the tone of this contribution should be lowered.\
(2) I am also skeptical of the main idea of employing the generative model's faulty region as pseudo-OOD data, which relies on the flaw of the diffusion model. Ideally, when the diffusion model can be able to generate in-distribution data perfectly, then what does the proposed algorithm of the paper become?\
(3) The paper's empirical performance is also weak. I can find several previously published works [2,3] that outperform the proposed method by a large amount. Furthermore, the proposed filtering strategy only improves by a small amount according to Table 3.\
(4) Since the method trains and generates data through the diffusion model, extra computational cost during training is non-negligible. Authors should specify such extra computation costs.\
(5) Back to the novelty, I can find similar work that also synthesizes outliers from the boundary or low-density regions [4,5]. I do not think applying a similar idea to the diffusion model is not strikingly novel or surprising.\
(6) While this idea can be extended to a classification setting, I highly doubt that this is an effective synthesis strategy compared to existing methods unless further experiments are done. Consider applying the algorithm in the classification setting.

Overall, I do not think the idea is relatively new or the performance of the paper is that significant. I am leaning toward rejecting the paper.

**Questions:**

See weakness

**References**\
[1] Fake It Till You Make It: Towards Accurate Near-Distribution Novelty Detection, ICLR 2023.\
[2] SAFE: Sensitivity-Aware Features for Out-of-Distribution Object Detection, ICCV 2023.\
[3] Deep Feature Deblurring Diffusion for Detecting Out-of-Distribution Objects, ICCV 2023.\
[4] Out-of-distribution Detection with Boundary Aware Learning, ECCV 2022.\
[5] Density-driven Regularization for Out-of-distribution Detection, NeurIPS 2022.

---

### Official Review · Reviewer_vpYS · 2023-10-29

**Soundness:** 3 good
**Presentation:** 3 good
**Contribution:** 2 fair
**Rating:** 5
**Confidence:** 4

**Summary:**

In this paper, the authors propose a novel approach for out-of-distribution (OOD) sample detection which leverages the diffusion model to generate data samples and incorporates a filtering mechanism to produce meaningful out-of-distribution (OOD) data. Furthermore, they propose a KNN-based filter to select OOD samples that not only have low probability density but also contribute to the training of the OOD detector. Different from other methods that rely on parametric models, the proposed diffusion model-based approach captures the latent feature distributions of ID data, thus allowing to synthesize OOD data samples within a compact feature space. The experimental validation is extensive and convincing.

**Strengths:**

The paper is well-documented, clearly written and easy to follow. The related work section covers most of the relevant articles in the related fields. The experimental validation is extensive. The proposed approach is conveniently compared with several state-of-the-art methods, demonstrating its superiority.

**Weaknesses:**

The paper seems to rely too heavily on the VOS paper (Du et al., 2022b), which makes questionable the novelty of the proposed approach.

**Questions:**

Here are my concerns:
- The authors have to position their approach with respect to the one described in the VOS paper and clearly discuss the difference between the two. It seems that the main difference relies on the usage of the diffusion models to generate the synthetic OOD samples and the KNN filter strategy to select low-density data samples as OOD samples. Please make more clear your contribution.
- It seems there is a confusion between outliers and OOD samples. Outliers belong to the seen classes (low density areas) whereas OOD samples belong to classes which have not been seen during training. It seems that your diffusion model generates outliers in low-density areas which are considered as OOD samples. Please clarify this aspect.
- Maybe I missed this point, but how do you define in your protocol which are ID and OOD classes? This question applies to all the datasets you considered for the experimental validation.

---

### Official Review · Reviewer_gKdQ · 2023-10-30

**Soundness:** 2 fair
**Presentation:** 2 fair
**Contribution:** 2 fair
**Rating:** 5
**Confidence:** 4

**Summary:**

The paper highlights a significant issue with current methods employed for generating out-of-distribution (OOD) feature samples. These methods rely on incorrect assumptions about the distribution of in-distribution (ID) data, resulting in limited performance improvement. To address this limitation, the paper introduces a novel approach that utilizes a diffusion model to generate OOD samples. Additionally, a filtering strategy based on K-nearest neighbors (KNN) is employed to identify and retain only those generated samples that belong to low-density regions of the ID data. These carefully filtered samples are then utilized to enhance the training of the model, leading to improved performance in OOD object detection.

**Strengths:**

1.	This paper leverages a diffusion model to generate OOD features without predefined data distribution.
2.	The paper is well-written and the methodology is clear and easy to understand.
3.	Extensive experiments are done to analyze various aspects of the method.

**Weaknesses:**

1.	The paper introduces a diffusion model to generate OOD features, which requires multi-time step inference, but the performance improvement it can bring is not obvious in the table 1 and 2.
2.	One of the main contributions highlighted in the paper is the utilization of K-nearest neighbors (KNN) to filter the generated data and obtain appropriate out-of-distribution (OOD) samples. However, it is worth noting that this approach has been previously proposed in [1]. Regrettably, this paper does not provide a direct comparison with the aforementioned work.
3.	The paper may not offer a significant level of innovation, and its potential to inspire other scholars might be limited.

[1] Tao L, Du X, Zhu J, et al. Non-parametric Outlier Synthesis[C]//The Eleventh International Conference on Learning Representations. 2022.

**Questions:**

see weakness

---

### Official Review · Reviewer_1VLd · 2023-10-30

**Soundness:** 2 fair
**Presentation:** 2 fair
**Contribution:** 2 fair
**Rating:** 5
**Confidence:** 3

**Summary:**

This paper presents a diffusion model-based data generation method for training detectors capable of detecting out-of-distribution (OOD) objects. The article utilizes the diffusion model to fit the latent feature distribution of intra-distributed (ID) data and designs a KNN-based filtering strategy to select low-density OOD samples. And experiments are conducted on several OOD detection datasets to demonstrate the effectiveness and superiority of its method.

**Strengths:**

1.	A novel data generation method is proposed for the challenging and important problem of OOD detection, which is innovative and valuable.
2.	The article uses a diffusion model to generate data without assuming any a priori distribution form, which is able to fit complex ID data distributions more accurately, and has advantages over traditional methods based on GMM or VAE, for example.
3.	A KNN-based filtering strategy is designed, which can effectively select low-density OOD samples close to the flow shape of ID data, improving the difficulty and effectiveness of OOD detection.
4.	The article conducts extensive experiments on several OOD detection datasets and compares them with a variety of benchmark methods to demonstrate the superiority and generalization of its approach.

**Weaknesses:**

1.	The writing quality of the article is poor, with some grammatical errors, spelling mistakes, punctuation errors and formatting errors, which affect the readability and professionalism of the article.
2.	The article does not give the specific details and parameter settings of the diffusion model and the filtering strategy.
3.	The article does not explore the effects of different modules or parameters on the performance of its methods.

**Questions:**

1.	On page 3, the definition of the diffusion model does not seem to be clear enough, please describe it in detail.
2.	The derivation of equation (2) on page 4 is not detailed enough and needs to be added
3.	How to choose the parameters of the filtering strategy mentioned in the article, especially the threshold value mentioned on page 5?
4.	The comparison with other methods in Figure 3 on page 7 is not intuitive enough, more comparison experiments are suggested to be added.
5.	The background introduction section is short, it is recommended to add an introduction to the work related to diffusion modeling.
6.	The article mentions "unknown distribution" several times, but does not give a clear definition, it is suggested to define it on page 9.
7.	The article mentions "prototype-based representation learning loss", but no specific definition or formula is given.
8.	Have the authors considered other alternatives to the filtering strategy mentioned in the article?

---

### Official Review · Reviewer_8gzR · 2023-10-31

**Soundness:** 3 good
**Presentation:** 3 good
**Contribution:** 2 fair
**Rating:** 5
**Confidence:** 3

**Summary:**

The paper proposes a new approach to address the OOD (Out-of-Distribution) problem. The author aims to enhance OOD detection by generating OOD samples. Specifically, the author suggests using a diffusion model to generate OOD samples. Additionally, the author introduces a new filtering method to select which generated samples should be considered as OOD samples. According to the author's experiments, their proposed approach can improve the effectiveness of OOD detection.

**Strengths:**

1. The approach proposed in this article is interesting, and there is currently limited exploration in this area. This article can provide new insights for addressing the OOD problem.
2. The experiments in this article are comprehensive, comparing various generation methods, including different datasets and approaches.
3. According to the author's experimental results, the proposed method in this article can improve the effectiveness of OOD detection.

**Weaknesses:**

1. The paper lacks clarity on how to choose which samples to consider as OOD samples. The author merely mentions using a filter to differentiate between in-distribution (ID) and OOD samples while avoiding samples that are too far from ID. Although this intuitively makes sense, the article lacks theoretical support or specific examples to illustrate what kind of samples would be considered ID or excessively far OOD.

2. In Table 2, the author compares the effectiveness of different generation methods, but from the results, diffusion does not seem to have a significant advantage. In general generation tasks, diffusion is usually considered superior to GANs. The relationship between the performance of the generation model and its improvement on OOD tasks is not analyzed.

3. The paper introduces a filter to distinguish between ID and OD based on a threshold. However, the selection and stability of this threshold are not discussed in detail. Additionally, there is no discussion on how to determine which samples are considered excessively far OOD samples using the filter.

**Questions:**

As shown in the weakness.